# Determinants of Soil Bacterial Diversity in a Black Soil Region in a Large-Scale Area

**Jiacheng Niu [1], Huaizhi Tang [1],\* , Qi Liu [1], Feng Cheng [2], Leina Zhang [2], Lingling Sang [2], Yuanfang Huang [1] , Chongyang Shen [1], Bingbo Gao [1] and Zibing Niu [1]**

[1] College of Land Science and Technology, China Agricultural University, Beijing 100193, China; jiachengniu@cau.edu.cn (J.N.); liuqi2021@cau.edu.cn (Q.L.); yfhuang@cau.edu.cn (Y.H.); chongyang.shen@cau.edu.cn (C.S.); gaobingbo@cau.edu.cn (B.G.); 2018321010103@cau.edu.cn (Z.N.)

[2] China Land Surveying and Planning Institute, Beijing 100032, China; chengfeng@mail.clspi.org.cn (F.C.); zhangleina@mail.clspi.org.cn (L.Z.); sanglingling@lcrc.org.cn (L.S.)

\* Correspondence: tanghz@cau.edu.cn

**Abstract:** Soils in black soil areas are high in organic matter and rich in nutrients. Soil microorganisms are particularly critical to cultivated land. The objective of this study was to explore the influencing factors of soil bacterial diversity under special regional conditions in a black soil region. In this study, the cultivated land in a black soil area was used as the study area and a random forest was used to map the bacterial abundance in the black soil area based on 1810 sample points. DbMEM analysis was used to quantify the spatial effect of the black soil area and to identify the influencing factors of soil bacterial abundance in the black soil area in combination with soil properties, terrain, and climate. Results of a variation division showed that broad (8.336%), AT (accumulated temperature, 5.520%), and pH (4.184%) were the main factors affecting soil bacterial diversity. The broad effect was more significant in the spatial effect, which may be related to the local landscape configuration. Overall, our research showed that the influencing factors of soil bacteria will be affected by regional characteristics.

**Keywords:** black soil area; soil bacterial diversity; influencing factors; spatial effect; spatial heterogeneity





## 1. Introduction

The black soil region of Northeast China has high soil fertility, it is suitable for cultivation, it has a high production potential, and it is an agriculturally productive soil resource in China [1]. The total area of arable land in the black soil region is about 50,000 km$^2$. In 2016, the total grain production in the black soil region of Northeast China was about 101 million tons, accounting for one-fifth of China's total grain output. However, due to instability of the environment and the continuous utilization of cultivated land, the soil in the black soil area has a number of problems, such as thinning of the soil layer, decreased fertility, and increased soil bulk density. In addition, erosion by local winds and water forces has caused rapid degradation of the cultivated land in the black soil area due to soil quality deterioration, affecting the sustainable development of agriculture in the region [2–4]. In this case, nutrient cycling, pollution purification, and ecological support dominated by soil microorganisms on cultivated land production and ecological functions are particularly important. Therefore, research on soil microbial diversity and its influencing factors in black soil areas is of great significance for the protection, quality restoration, and improvement of cultivated land in black soil areas.

For decades, several studies have focused on soil microorganisms and soil properties [5–7], climate [8,9], vegetation [10,11], and land use [12,13]. Soil properties, represented by pH, are often the primary factor affecting soil microorganisms. Other factors, such as climate and terrain, have a smaller impact [14,15]. Climate and terrain often affect

soil microorganisms by changing soil properties [16,17]. However, it is not clear whether special regional characteristics, such as areas with significant climate change, will affect soil microorganisms [18].

In recent years, some studies have compared the influencing factors of soil microbes at a global scale [19,20], national scale [21,22], and regional scale [23,24]. These works found that environmental changes due to geographic distance alter microbial community composition in different regions and affirmed the important influence of spatial factors on soil microorganisms. However, in these large-scale studies, there was often spatial heterogeneity in different partitions of the same region. Spatial heterogeneity leads to different values of changes in soil properties, climate, and terrain in different regions; thus, it further intensifies the contradiction of soil microbial influencing factors and affects the ecological processes (deterministic vs. neutral processes) of soil microorganisms [14,19]. Studying the ecological processes of soil microorganisms is helpful to better protect and utilize soil microorganisms. However, quantifying the spatial effect by geographical distance cannot analyze the ecological processes of soil microorganisms [25].

The purpose of this study was to investigate the influencing factors of soil bacterial diversity in a large-scale area, such as the black soil region, where soil organic matter content is high and nutrient-rich; the terrain chosen was flat and the soil conditions had rapidly degraded. This study used a random forest to predict and map soil bacterial diversity in the black soil area. It quantified the spatial effect of the black soil area using the dbMEM method and determined important factors affecting the soil bacterial diversity of the cultivated land in the black soil area in combination with environmental variables, such as soil properties, climate, and terrain.

## 2. Materials and Methods

### 2.1. Study Area

In this study, the black soil region in Northeast China was selected as the study area (115° E–135° E, 48° N–55° N) which has a flat terrain and rich soil organic matter. Most of the area is located in the middle temperate zone with a small part in the cold temperate zone and the warm temperate zone. The winter is cold and long and the summer is warm and short. The annual average temperature ranges from –4 °C to 10 °C. The rainfall is concentrated in the form of heavy rain, which falls in July to September, accounting for about 70% of the annual rainfall with an average of 400–700 mm. A total of 1878 sample points were selected, which sampled the agricultural land in this area. Each county-level administrative unit has five sample points, which were selected from farmland integrating factors, such as land use type in the county. The sample points were selected as follows: black and dark brown soil areas in the low foothills of the Xiaoxing'an Mountains (I, 212 sample points), brown and black soil areas in the Liaohe Plain (II, 326 sample points), white mud soil areas in the Sanjiang Plain (III, 234 sample points), black soil calcareous region of the Songnen Plain (IV, 359 sample points), western semi-arid area (V, 233 sample points), and dark brown soil area in the low hilly area of Changbai Mountain (VI, 514 sample points). The regions are shown in Figure 1B.

### 2.2. Data Sources

2.2.1. Soil Bacterial Abundance Data

This study took the black soil area of China as the study area and conducted a study based on the soil bacterial abundance index data from the "Third Land and Resources Survey in China" [26]. This survey began in September 2018 and ended on 31 December 2019. This work had comprehensively investigated the utilization of China's land resources and established a land survey database covering national, provincial, prefectural, and county levels. In a special investigation of cultivated land resources, the quality classification of cultivated land resources includes 10 indicators, such as soil thickness, soil pH, organic matter content, soil microbial diversity, etc. [26]. Soil bacterial abundance sampling was characterized using the Chao1 index. The Chao1 index refers to the index for estimating

the number of soil bacterial OTUs contained in a sample using the Chao1 algorithm [27]. It is commonly used in ecology to estimate total number of species. The Chao1 index is one of the Alpha diversity indices. Other Alpha diversity indices include the Ace index, Simpson index, Shannon index, etc. Data details are shown in Figure 2.

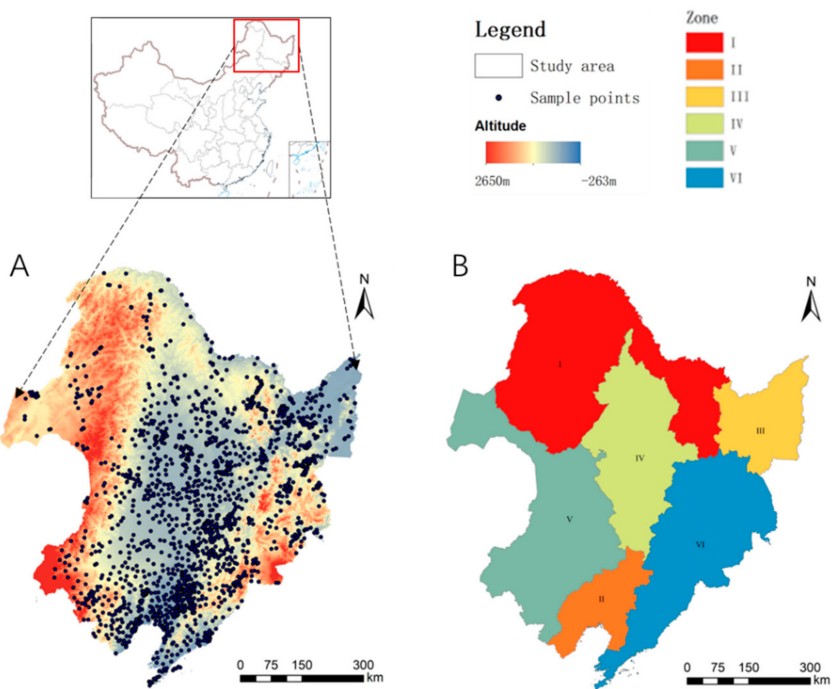

**Figure 1.** Study area. (**A**) Location of sampling points in the black soil region. (**B**) Distribution of the black soil region partitions; partitions were mainly based on soil types.

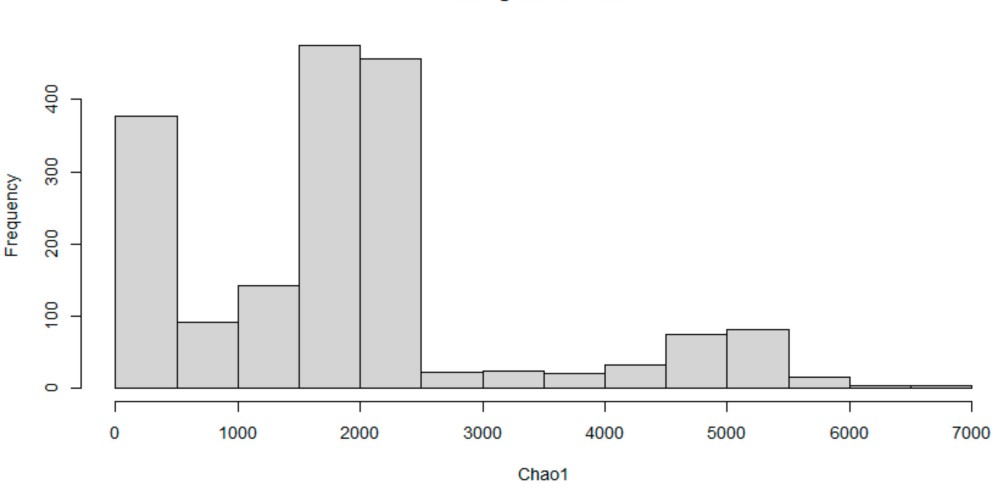

**Figure 2.** Chao1 index histogram.

### 2.2.2. Environmental Variables Data

SOM (soil organic matter) and pH data come from the "Third Land and Resources Survey in China" (2021). AT (accumulated temperature) data and MAP (mean annual precipitation) come from the "Space of the Annual Average Temperature in China since 1980 Interpolation Dataset" (2015). Altitude and slope data are from the "China's Altitude (DEM) Spatial Distribution Data" (2000). BD (bulk density) and CEC (cation exchange capacity) data are from the "Soil Map-based Harmonized World Soil Database (v1.2)" (1995). ClayPerc and SandPerc data are from the "Spatial Distribution Data of Soil Texture

in China" (2019). We combined and cut the environmental variables data according to the study area for display purposes. Figure 3 shows details of the environment variables. Details of environmental variables for each zone are shown in Table 1.

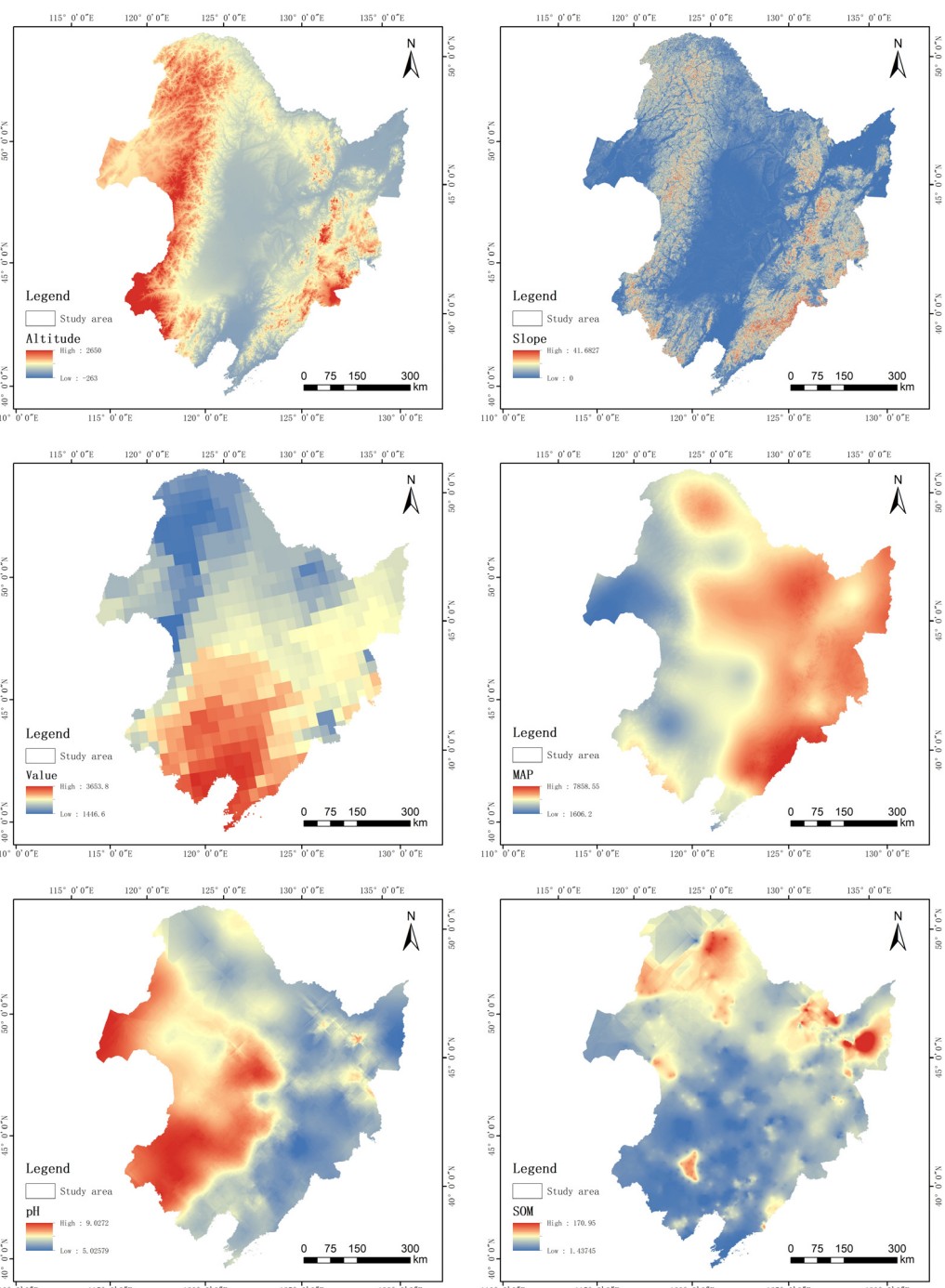

**Figure 3.** *Cont.*

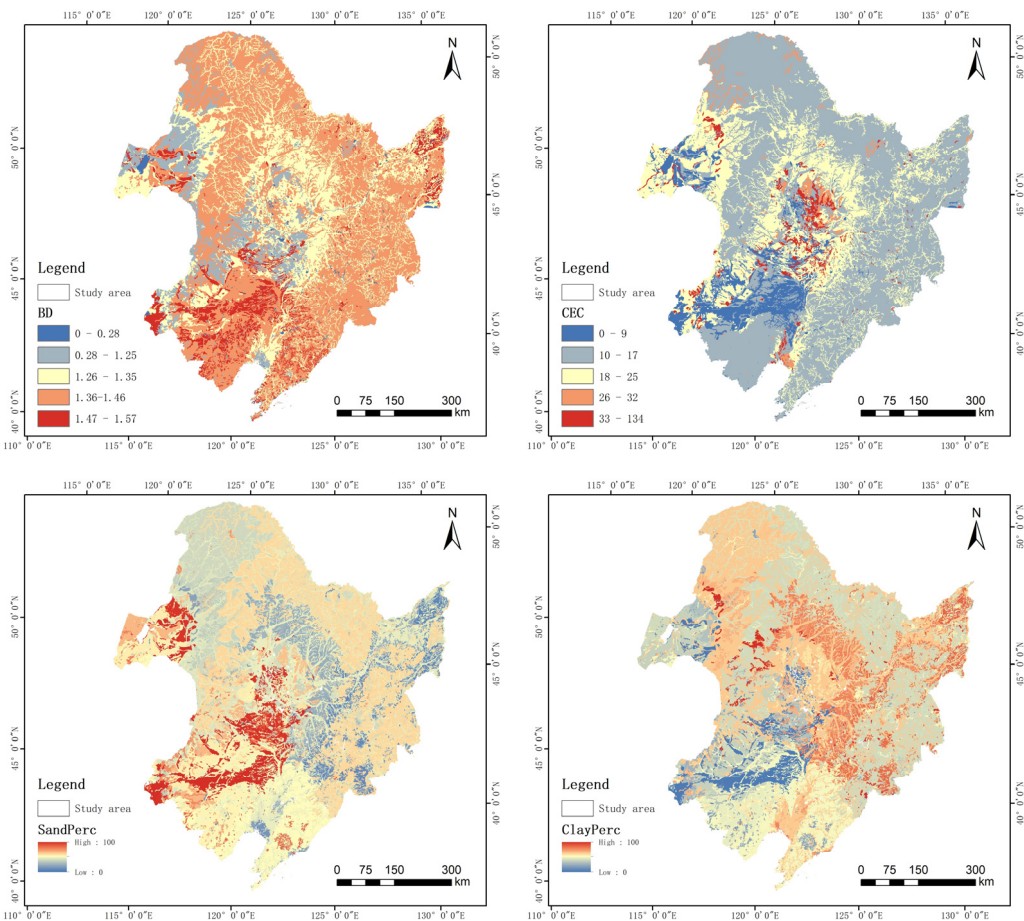

**Figure 3.** Spatial distribution of environmental variables.

**Table 1.** Mean value of environmental variables for each zone.

| Variables | I | II | III | IV | V | VI |
|---|---|---|---|---|---|---|
| AT (°C) | 1955.82 | 3476.52 | 2380.96 | 2499.56 | 2805.46 | 2752.80 |
| MAP (10 mm) | 5225.00 | 4853.77 | 5575.94 | 5150.50 | 3772.33 | 5862.68 |
| Altitude (m) | 381.07 | 70.71 | 149.78 | 173.54 | 434.36 | 249.46 |
| Slope (°) | 1.20 | 0.65 | 1.23 | 0.47 | 1.06 | 1.83 |
| pH | 6.50 | 6.33 | 6.48 | 7.22 | 8.12 | 6.03 |
| SOM (g·Kg$^{-1}$) | 57.05 | 26.47 | 50.59 | 30.45 | 26.10 | 32.89 |
| BD (kg/dm$^3$) | 1.36 | 1.39 | 1.38 | 1.38 | 1.44 | 1.39 |
| CEC (cmol/kg) | 19.28 | 17.22 | 16.48 | 20.73 | 15.89 | 16.31 |
| ClayPerc (%) | 23.92 | 24.44 | 26.42 | 26.50 | 20.27 | 26.09 |
| SandPerc (%) | 46.90 | 44.99 | 41.53 | 46.43 | 54.24 | 26.09 |

### 2.3. Data Processing

First, we used ordinary Kriging interpolation for pH and SOM data based on points; parameters were automatically fitted by Arcgis 10.7. The slope data were calculated based on DEM data of the black soil area. AT data were converted from vector to grid; other data were from raster datasets. After processing, all data were in a kilometer grid. Based on the study area, the data of all environmental variables were combined, cut, and displayed by mapping (Figure 3). Then, ten indicators from each dataset were extracted based on the sample points from each dataset including climate (AT, MAP), terrain (altitude, slope), and soil properties (pH, SOM, SandPerc, ClayPerc, BD, and CEC). Each null index was replaced by the average value of the index that removed the nulls. Outliers and extreme values were removed from the data; a total of 68 points were removed and 1810 points remained. The R 4.4.1 ("car" package) was used to eliminate a variance inflation factor (VIF) greater than five in the environmental variables in order to eliminate the collinearity between each

factor and the Chao1 value. After screening, the VIF of each index was less than five and the ten indexes were modeled using the "lme4" and "tidyverse" packages in R. The spatial prediction data were based on the processed data of the mixed model and the influencing factor analysis was based on the original data.

*2.4. Research Method*

2.4.1. Random Forest

Random forest is an ensemble method that combines multiple regression trees (classification or regression) to give predictions [28]. The distribution of each tree is random and depends on the value of a random vector, which is sampled independently [28]. A regression tree is an algorithm that involves recursively splitting data into several simple regions using a series of splitting rules. It is called a decision tree because the series of splitting rules can be summarized as an inverted tree structure [29]. The generalization error of RF is based on the strength and correlation of each decision tree and the result is the average set of multiple decision trees. Therefore, RF has more stable denoising ability.

In addition to the input variables, the stochastic forest model has the two following parameters: $n_{tree}$ and $m_{try}$. The $n_{tree}$ refers to the number of trees used in RF, which determines the complexity and accuracy of RF [28]. The $m_{try}$ determines the strength and relevance of each decision tree in RF. The higher the $m_{try}$, the higher the strength and correlation between trees, but the performance will decline [30,31].

In this study, the "randomforest" and "rfpermute" packages were used for simulation modeling and the optimal parameters were selected according to $R^2$ [32]. Then, random forest modeling and importance factor ranking were carried out; the training set (80%) and test set (20%) were divided for accuracy verification.

2.4.2. Variation Partition

We used the "Vegan" package to estimate the effects of soil physicochemical parameters, climatic conditions, terrain, and spatial effects on soil bacterial abundance through variance partitioning. First, the environmental variables were grouped. The groupings were divided as follows: AT and MAP indicators into climate; altitude and slope into the terrain; the remaining pH, SOM, and other indicators into soil property; the selected dbMEM variables into space descriptors. Then, the intra- and inter-group explanation rates were calculated and forward and backward selections were performed to screen out the significant environmental variables involved in the variation partition [33]. Finally, the "ggplot2" package was used to map in R.

2.4.3. DbMEM

The Moran's eigenvector map (MEM) method can generate $n-1$ spatial variables with positive or negative eigenroots to obtain spatial variables that simulate positive and negative spatial correlations in a larger range [25]. The MEM eigenvector maximizes the Moran exponent, which equals the Moran exponent multiplied by a constant. The Moran index is calculated as follows:

$$I(x) = \frac{\frac{1}{w} \sum_{h=1}^{n} \sum_{i=1}^{n} w_{hi}(y_h - \overline{y})(y_i - \overline{y})}{\frac{1}{n} \sum_{i=1}^{n}(y_i - \overline{y})^2} \tag{1}$$

where $W_{hi}$ is the weight of matrix $W$ and $y_h$ and $y_i$ are the values of variable $y$ in quadrature h and quadrat $i$. The dbMEM used in this study is a special case of MEM, also known as the principal coordinate of the neighbor matrix (PCNM) method. After calculating the Moran eigenvectors, the method tends to retain the positive spatially correlated eigenvectors; the PCNM eigenvectors and spatial correlation coefficients generated in dbMEM are often not linear.

In this work, dbMEM analysis was performed using the "adespatial" package and a total of 907 dbMEMs were fitted [34]. Forward selection was performed on all dbMEMs to

obtain 63 significant dbMEMs, which were further screened (27) and divided according to $R^2$. For the three spatial scales (broad, medium, and fine), the dbMEM with the highest $R^2$ was selected from each scale to participate in the variance partition.

## 3. Results

### 3.1. Soil Bacterial Abundance Prediction Mapping

We removed outliers from the 1878 sampling points in the original data; there were 1810 remaining sampling points. Random forest modeling was performed using the processed data and ten indicators were selected from the three following factors: climate, terrain, and soil properties. A subset of random forests was divided at the time of prediction, 80% was used for training sets (1447) and 20% for test sets (363) in subsequent accuracy assessments.

Ten environmental variables were input into the model for debugging and predictions. After debugging, the random forest parameter $n_{tree}$ was 100 and the model explanation value ($R^2$) was 0.970. The predicted values of the random forest model ranged from 1705.55 to 2574.34, with an average of 2179.64 and a standard deviation of 171.00. Among the accuracy test indicators, ME (mean error) was 2.577; RS (residual error) was 22.274 and RSME (root mean square error) was 35.084.

The forecast mapping of the northeastern black soil area using ordinary kriging is shown in Figure 4. The optimal fitting model of the semi-variance function performed by GS+ was a Gaussian model and the block-to-basic ratio was 14.63% (<25%), indicating that the soil bacterial abundance in the black soil area had a strong spatial autocorrelation with an $R^2$ of 0.971. Figure 4 shows that the soil bacterial abundance decreased gradually from south to north.

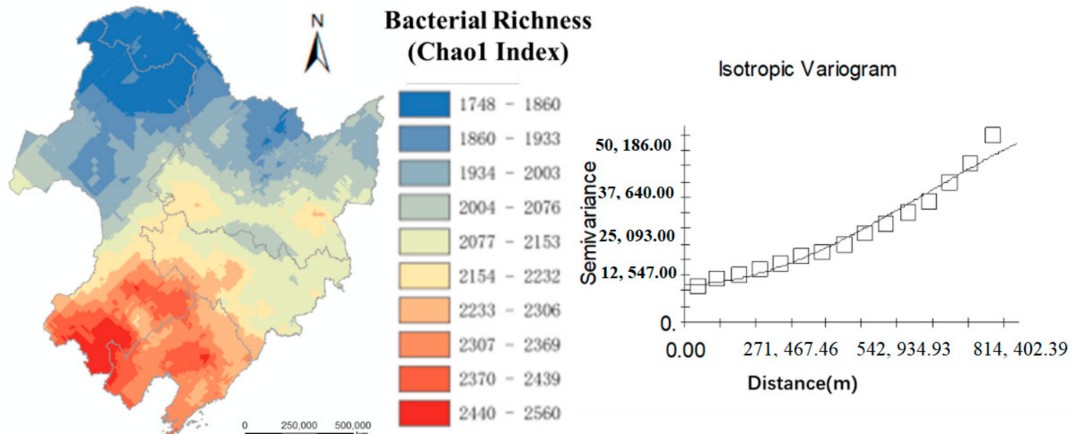

**Figure 4.** Spatial distribution of the Chao1 index in the black soil area. The colors indicate the extrapolated values expressed as Chao1 per soil sample. In the graph, the points represent the experimental variogram and the continuous lines are the Gaussian models fitted by GS+.

### 3.2. Soil Bacterial Abundance and Environmental Variables

We performed forward selection on all indicators and screened out significant environmental variables for variation partitioning to measure the individual and combined effects of soil properties, climate, terrain, and spatial effects. The explained variance of each index to soil bacterial abundance was 68% (Figure 5) and the order of the individual interpretation rates of each part was as follows: spatial effect (9%) > climate (6%) > soil property (5%) > terrain (3%). The results of Pearson correlation analysis showed the top five indicators with the strongest correlation with soil bacterial abundance as follows: MEM1 (r = 0.673, $p < 0.05$) > altitude (r = 0.601, $p < 0.05$) > pH (r = 0.487, $p < 0.05$) > AT (r = −0.431, $p < 0.05$) > MAP (r = −0.362, $p < 0.05$). Results are shown in Table 2.

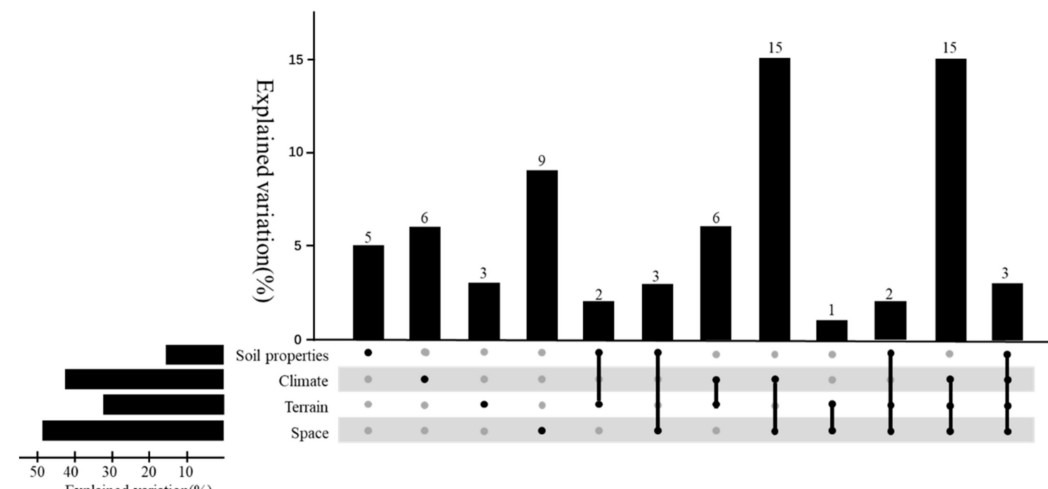

**Figure 5.** Variation partition. An UpSet plot showing the results from variation partitioning models that were used to identify the effects of soil properties, climate, and terrain and spatial descriptors. The dots below the bar represents the combination of different environmental variables. The black band represents explained variance of different environmental variables on soil bacterial abundance. The total explained variance of each part, such as the climate part, is shown on the left. Soil properties are pH, BD, CEC, and ClayPerc; climate is AT; terrain is altitude; spatial descriptors are MEM1, and MEM22, and MEM38.

**Table 2.** Analysis of influencing factors of Chao1.These value are based on the 1810 sites analyzed. The impact of each indicator on soil bacteria is evaluated by Pearson test. Partial indicators are screened to participate in the variation partition.

| | Chao1 | |
|---|---|---|
| | **Pearson Coefficient** | **Explained Variance (%)** |
| **Soil properties** | | |
| pH | 0.487 ** | 4.184 |
| SOM | 0.123 ** | |
| BD | 0.149 ** | 0.723 |
| CEC | −0.011 | 0.000 |
| ClayPerc | −0.124 ** | 0.000 |
| SandPerc | 0.150 ** | |
| **Climate** | | |
| AT | −0.431 ** | 5.520 |
| MAP | −0.362 ** | |
| **Terrain** | | |
| Altitude | 0.601 ** | 3.169 |
| Slope | 0.002 | 0.000 |
| **Spatial descriptors** | | |
| Broad [148 km,222 km] | | |
| MEM1 | 0.673 ** | 8.336 |
| Medium [74 km,148 km] | | |
| MEM22 | 0.128 ** | 0.443 |
| Fine [30 km,74 km] | | |
| MEM38 | 0.059 | 0.368 |

PS: $p < 0.05$: *; $p < 0.01$: **, $p < 0.001$: ***.

In parts of the soil properties, we screened indicators, such as pH, BD, CEC, and ClayPerc, to participate in the variation partition, shown in Table 2. The explained variance of the pH single indicator was 4.184% and the BD was 0.723%. The soil properties were mainly related to terrain and spatial effects and were combined (the explainability ranged from 2% to 3%); the soil property indicators were significantly correlated with the soil

bacterial abundance ($p < 0.05$) and the correlations of other indicators were weak, except for pH. Among the climate indicators, we chose AT to participate in the variation partition. The explained variance of the AT index was 5.520% and the co-explainability of climate and terrain, climate and spatial descriptors, and their common interpretations were 6%, 15%, and 15%, respectively; AT and MAP were negatively correlated with soil bacterial abundances. For the terrain, the explained variance of altitude was 3.169%. Interaction between terrain and spatial descriptors was 1%. Altitude had a significant correlation with soil bacterial abundance (r = 0.601, $p < 0.05$).

### 3.3. Soil Bacterial Abundance and Spatial Effect

We performed spatial effect fitting according to the study area and a total of 27 significant dbMEMs were fitted, shown in Figure 6A, each representing a different spatial scale (broad: 148~222 km, medium: 74~148 km, fine: 30~74 km, Table 2). We screened the most significant dbMEM for $R^2$, shown in Figure 6B, involved in the variation partitioning at each scale. The order of interpretation of dbMEM at different scales was as follows: broad (8.336%) > medium (0.443%) > fine (0.368%). The Pearson correlation coefficient also showed the same trend of broad (r = 0.673, $p < 0.05$) > medium (r = 0.128, $p < 0.05$) > fine (r = 0.059). The analysis results of the influencing factors at different scales showed the following: the broad scale was more correlated with AT, MAP, and altitude indicators ($p < 0.01$); the medium was correlated with other indicators except for slope ($p < 0.05$); the fine was correlated with BD, CEC, AT, and MAP ($p < 0.05$). The results are shown in Figure 6C.

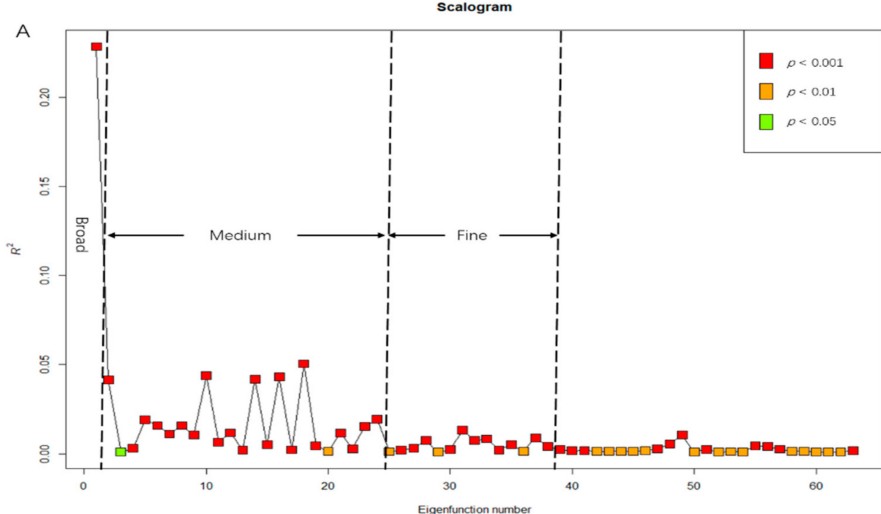

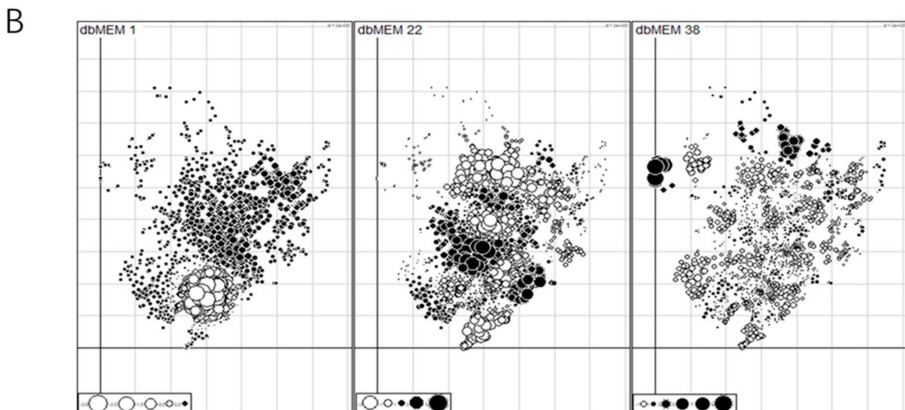

**Figure 6.** *Cont.*

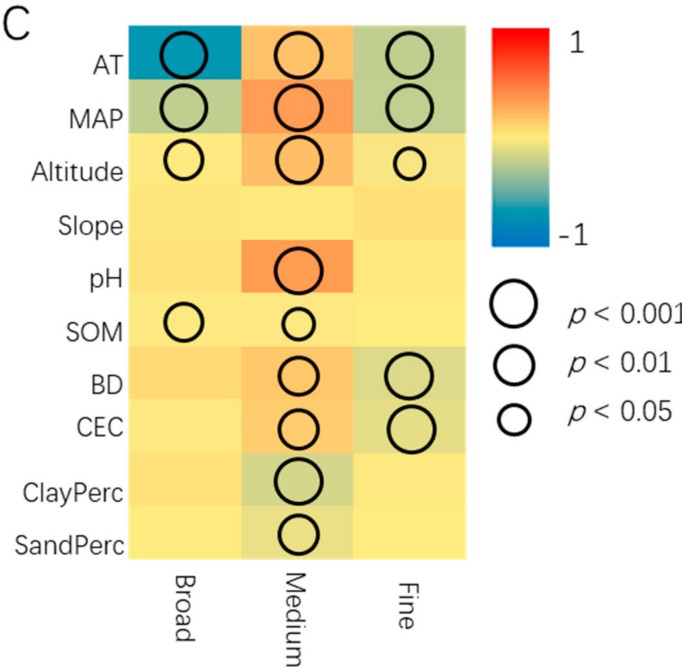

**Figure 6.** (**A**) Scale division of 63 significant dbMEMs, divided into broad, medium, fine; (**B**) the dbMEM display graph with the highest R² screened out from three different scales; (**C**) factor analysis graph based on the screened dbMEM. The squares in (**A**) represent dbMEM and the colors represent significance. (**B**) shows dbMEM, fitted at different scales; the circles represent the spatial structure and the colors represent positive and negative correlations.

### 3.4. Soil Bacterial Abundance under Different Zoning

In order to eliminate the influence of the spatial effect, further specific influencing factors need to be found. We used variation partition and Pearson analysis to analyze the influencing factors of soil bacterial abundance in different zones, shown in Figure 1B. We used all variables to participate in the calculation. The results of the variation partition in different zones are as follows: VI (59%) > I (58%) > V (31%) > III (22%) > II (11%) > IV (8%). In zone I, the order of explained variance of each influencing factor was as follows: soil properties (21%) > interaction (18%) > terrain (17%) > climate (3%). In zone II, the order of explained variance of each influencing factor was as follows: climate (8%) > soil properties (3%). In zone III, the order of explained variance of each influencing factor was as follows: interaction (9%) > terrain (6%) > climate (4%) > soil properties (3%). In zone IV, the order of explained variance of each influencing factor was as follows: climate (6%) > terrain (2%). In zone V, the order of explained variance of each influencing factor was as follows: soil properties (25%) > climate (6%). In zone VI, the order of explained variance of each influencing factor was as follows: interaction (30%) > climate (25%) > soil properties (2%) = terrain (2%).

The results of the Pearson analysis showed that different zones had different influencing factors. In zone I, soil bacterial abundance was significantly correlated with MAP (r = −0.634, $p < 0.01$), pH (r = 0.628, $p < 0.01$), and altitude (r = 0.586, $p < 0.01$). In zone II, soil bacterial abundance was significantly correlated with AT (r = −0.246, $p < 0.01$), altitude (r = 0.169, $p < 0.01$), and MAP (r = −0.167, $p < 0.01$). In zone III, soil bacterial abundance was significantly correlated with altitude (r = 0.325, $p < 0.01$), MAP (r = 0.290, $p < 0.01$), and slope (r = 0.242, $p < 0.01$). In zone IV, soil bacterial abundance was significantly correlated with SOM (r = 0.288, $p < 0.01$), AT (r = −0.241, $p < 0.01$), and slope (r = 0.170, $p < 0.01$). In zone V, soil bacterial abundance was significantly correlated with pH (r = 0.379, $p < 0.01$), altitude (r = 0.244, $p < 0.01$), MAP (r = −0.232, $p < 0.01$), and SOM (r = 0.199, $p < 0.01$). In zone VI, soil bacterial abundance was significantly correlated with AT (r = −0.714, $p < 0.01$), altitude (r = 0.531, $p < 0.01$), and SOM (r = 0.202, $p < 0.01$).

## 4. Discussion

Although several studies have compared macroscopic factors, such as climate and terrain, soil properties have a more important impact on soil bacteria. However, few studies have compared the influencing factors of soil bacteria under different soil conditions. In this study, we selected a black soil area with rich soil fertility, organic matter content, and nutrients as the study area to evaluate whether the influencing factors of soil microorganisms are consistent under different soil conditions. Our research showed that the influencing factors of soil bacteria in the black soil area were special. In the black soil area, due to favorable soil conditions, the influence of the soil properties on soil bacterial abundance is reduced, while macroscopic factors, such as climate, elevation, and spatial effects, will have a stronger impact. Environmental filtration is the main ecological process for the construction of local microbial communities. More importantly, we found that soil bacteria in different regions of the black soil area have different influencing factors, which indicates that it is necessary to consider spatial heterogeneity and the characteristics of different regions when conducting regional and even larger-scale studies in the future.

### 4.1. Particularity of Influencing Factors of Soil Bacterial Abundance in Black Soil Region

In this study, we used variation division to analyze the main factors affecting soil bacteria. The contribution results showed that accumulated temperature (5.520%) and altitude (3.169%) in the black soil area were more significant factors than soil attributes (pH, 4.184%; BD, 0.723%), which was different from the research conclusions of Liu et al. [35] and Wang et al. [36]. Delgado Baquerizo et al. found that soil properties, such as pH and soil carbon content, are more important after studying global-scale soil bacterial diversity and its environmental impact factors [19]. Similar results were obtained by Rousk et al. [16] and Xue Peipei et al. [23]. Considering regional factors, we selected studies at similar latitudes for comparison. Relevant studies from Griffiths et al. in the UK [20] and Terra et al. in France [37] also indicate that soil pH has the most significant effect on soil bacteria. Compared with the soil pH range in France (3.70–8.90) [37], the soil pH in black soil area was higher (4.65–9.25), as well as their mean value was 6.42, which was close to our mean value of 6.68. Therefore, the soil pH in the black soil area is in the normal range. Some scholars even believe that macro factors, such as climate and landform, barely participate in the construction of microbial abundance and diversity distribution patterns [38,39]; soil attributes are often considered to be the primary factor affecting soil microorganisms (contribution is 18–40%).

The results of this study showed that macro factors, such as climate and terrain, in black soil areas have a more significant impact on soil bacteria and that soil bacteria show a certain gradient under the conditions of accumulated temperature and altitude change (Figure 7A), but the gradient is not obvious under soil conditions (Figure 7B). We analyzed the climatic conditions in the black soil region of China and found that there is particularity in the distribution of the climate belt, showing a distribution that is perpendicular to latitude [40]. The climate and soil conditions in the black soil area are consistent (Figure 3) and the distribution pattern of the accumulated temperature and SOM is somewhat similar. The distribution pattern of precipitation and pH is similar, which may be because the local macro climate controls the distribution of soil conditions and affects the distribution of the soil bacterial community [8,41]. Other scholars have proved that rapidly changing environmental variables, such as available phosphorus, nitrogen, and other soil nutrient indicators, will drive the change of the soil microbial composition [42], which is closely related to local land-use and management methods. However, most of the environmental indicators selected in this study are relatively stable indicators and land-use indicators are not included in the study, which may be one of the reasons why the soil properties are not significant.

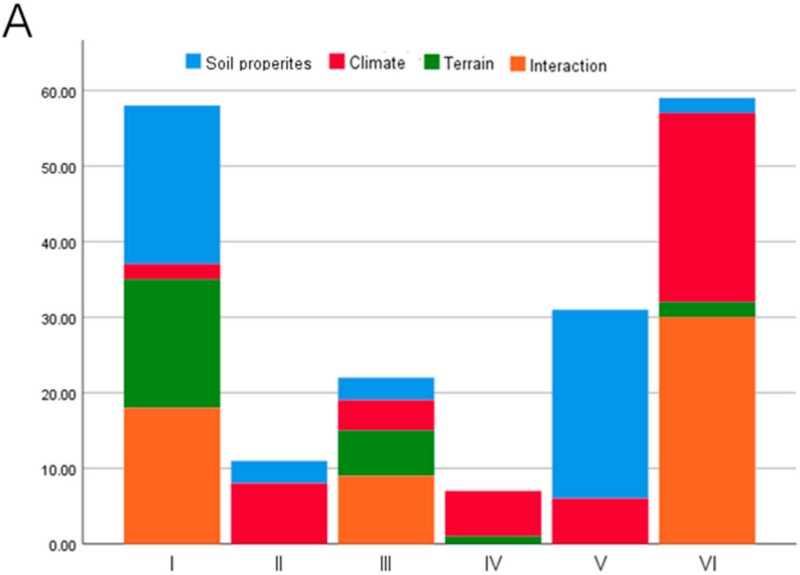

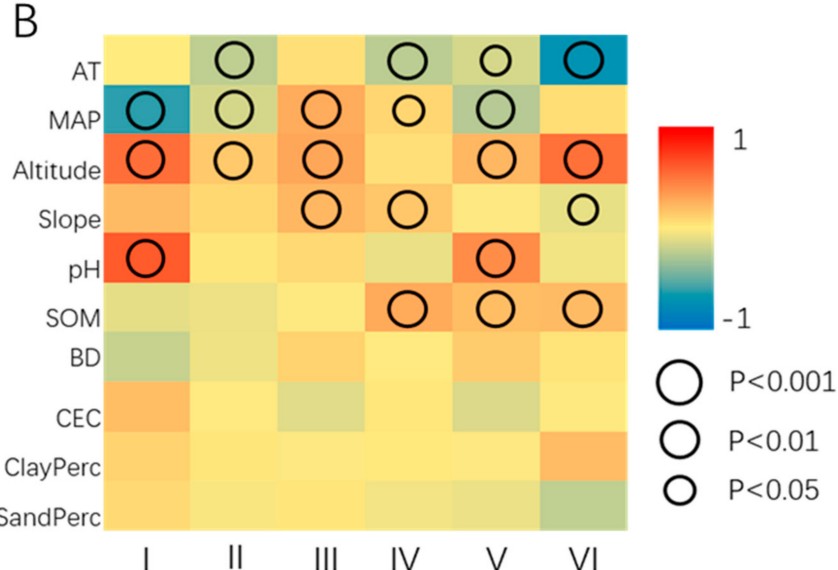

**Figure 7.** Analysis of bacterial abundance in partitions. (**A**) Variation partition of partitions, which includes soil properties, climate, terrain, and interaction. (**B**) Pearson coefficient of partitions; colors indicate the value of the Pearson coefficient and the circles indicate the confidence level. I, the black and dark brown soil areas in the low foothills of the Xiaoxing'an Mountains. II, the brown and black soil area in the Liaohe Plain. III, the white mud soil area in the Sanjiang Plain. IV, the black soil calcareous region of Songnen Plain. V, the western semi-arid area. VI, the dark brown soil area in low hilly area of Changbai Mountain.

*4.2. The Effect of Broad Scale on Soil Bacteria Is More Significant Than That of Medium and Fine Scale*

Through dbMEM analysis, we quantified the spatial effect of the black soil area and performed division of variation. The results showed that the spatial effect (about 50%) has a significant impact on soil bacteria, which was consistent with the research conclusions of Fierer et al. [17] and Zhang et al. [42]. Without considering the spatial effect, the result of the variation division was only about 30%. After adding the spatial effect, the interpretation degree of variation division increased by 15–25% [35,43].

Our results showed that the spatial effects of the medium and broad scales (8.336%) have a stronger impact on soil bacteria than those of the medium (0.443%) and fine scales

(0.368%). Terra et al. quantified the influence of the spatial effect on soil bacterial abundance through MEM analysis in France. In their research results, the microscale effect is dominant in the spatial effect of soil bacteria in France, which may be related to the fact that soil pH value is the primary influencing factor of soil bacteria in France [33]. In this study, the primary influencing factors of soil bacteria were climate and terrain, so the broad-scale effect was more significant. Spatial effects (MEM analysis) are often used to explain biological ecological processes [44]. In this study, the neutral processes represented by the microscale effect (drift, random diffusion, etc.) accounted for only a small part of the ecological process. Environmental filtration, dominated by a broad-scale effect, was the main ecological process of soil bacteria in the black soil area. Scholars have proved that the landscape configuration around cultivated land can affect the ecological processes of soil microorganisms and the landscape configuration can provide a source and sink for the migration of soil microorganisms, thus, affecting the ecological processes of soil bacteria [45,46]. However, it is not clear which is more important for soil microorganisms, macro factors or landscape configuration.

### 4.3. Influencing Factors of Soil Bacterial Abundance in Different Partitions

We conducted variance zoning and Pearson correlation analysis on different zones, as shown in Figure 8 The research showed that climate, terrain, and altitude still have significant effects on soil bacteria, but soil bacteria in different zones have different influencing factors, which may be related to spatial heterogeneity. Spatial heterogeneity leads to changes in soil types, land use, and soil properties and affects the distribution of soil microorganisms [45,47]. Compared with plain cultivated land (II, III, IV, V), mountainous cultivated land (I, VI) has a higher degree of interpretation, which may be due to large changes in soil conditions, climate, and vegetation caused by altitude, which affects the soil bacterial community [48]. Therefore, area I and VI are more significantly affected by climate and topography. Plain areas have more complex utilization methods [13] and landscape configurations [49–51]. The influencing factors of soil bacteria are more diverse and the influencing mechanism is more complex. Therefore, the influencing factors in plain areas were not significant. In addition, the influencing factors of soil bacteria selected in this study were relatively simple. In the future, when studying soil microorganisms in plain areas, we should select as many influencing factor indicators as possible and then find the important factors affecting soil microorganisms in plain areas.

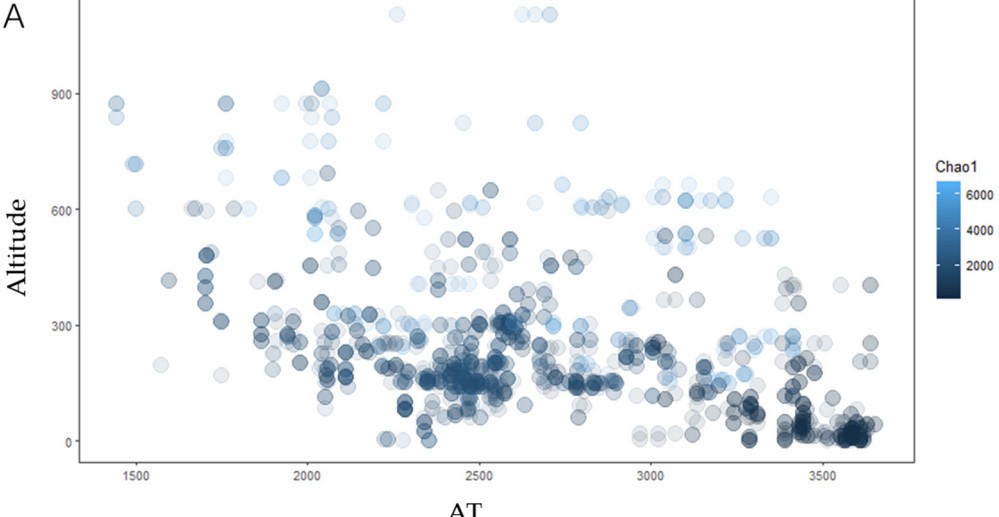

**Figure 8.** *Cont.*

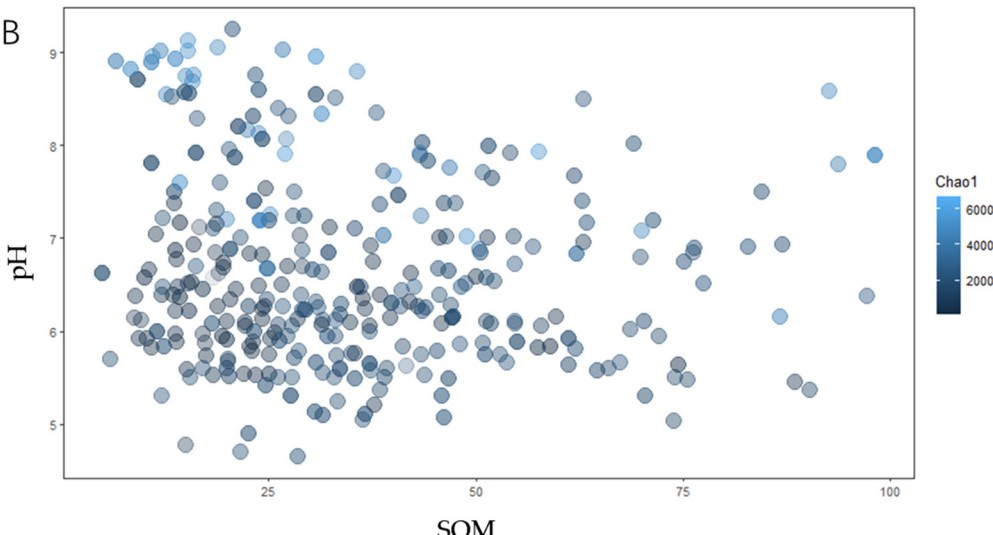

**Figure 8.** Comparison of the distribution of soil bacterial abundance under the conditions of altitude and accumulated temperature (**A**), and pH and soil organic matter (**B**). The darker the color of bubbles, the higher the Chao1 index. (**A**) Altitude (m), AT (°C); (**B**) SOM (g/kg).

## 5. Conclusions

In conclusion, our study provides a map of soil bacterial abundance distribution in the black soil region of Northeastern China. It quantified the spatial effect of soil bacterial abundance in the cultivated land in the black soil region using dbMEM analysis and identified the influencing factors of soil bacterial abundance. This study found the following: (1) broad (8.336%), AT (5.520%), and pH (4.184%) were the main factors affecting the distribution of soil bacterial abundance in the black soil area; (2) in terms of spatial effects, broad scale (148–222 km, 8.336%) > medium (74–148 km, 0.443%) > fine (30–74 km, 0.368%); (3) the influencing factors of soil bacteria in different regions were not consistent and the cultivated land in mountainous areas had a higher degree of explanation than that in plain areas. Our research shows that the influencing factors of soil bacteria are related to regional characteristics. Therefore, when considering the differences in cultivated land quality in different regions, we should fully consider the regional spatial heterogeneity and landscape configuration, which are closely related to human activities, in order to provide a good method for microbial migration and diffusion. This will allow for more scientific and rational use, better management of cultivated lands, and improvement of the productivity of cultivated lands.

**Author Contributions:** Writing—original draft, J.N. and H.T.; Data collection, collation and analysis, F.C., L.Z., L.S. and B.G.; data processing, Q.L. and Z.N.; methodology and research ideas, Y.H. and C.S. All authors have read and agreed to the published version of the manuscript.

**Funding:** This research was funded by the National Key R&D Program of China (Grant No. 2021YFD1500201.), the Young Scientists Fund of the National Natural Science Foundation of China (Grant No. 69191019), the Government Fund of the Land Renovation Center of Ministry of Natural Resources (Grant No. 2021-08-05).

**Institutional Review Board Statement:** Not applicable.

**Informed Consent Statement:** Not applicable.

**Data Availability Statement:** All environmental used in the study are available for free download through https://www.resdc.cn/Default.aspx (accessed on 15 March 2022) and http://www.tpdc. ac.cn/zh-hans/ (accessed on 17 March 2022). The soil bacterial abundance data are not publicly available due to legal requirements.

**Conflicts of Interest:** The authors declare no conflict of interest.

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
