# Peer review of "Determinants of Soil Bacterial Diversity in a Black Soil Region in a Large-Scale Area"

_land, doi:10.3390/land11050731_

Round 1
Reviewer 1 Report
Line 40 – numerous studies – only a few are specified. I am suggesting the usage of “several studies”
Lines 59- 61 – please move the phrase to materials and methods
Lines 64-68 – please move the phrase to materials and methods
Line 89 - Chao1 89 index – there are no references specified
Line 95 - Third Land and Resources Survey in China - there are no references specified
Line 102 – 105 - there is no references specified
Line 109 – figure 3 – is made by authors or is taken from elsewhere? Please specify the source.
Line 126 - Random forest – no reference provided
Lines 131-133 - no reference provided
Lines 135-143 - no reference provided
Discussions – difficult to follow. Please check the English translation. There are repetitive words that make the reading even more difficult (i.e. this study, scholars)
Author Response
Thank you for your comments, whcihe have uploaded in word.

Reviewer 2 Report
- It is unclear how pH and CTC can influence bacterial diversity and how these variables differ across areas. Please check this.
- Please submit data on soil attributes
- The authors do not submit the climatic characteristics of the studied areas
- The influence of climatic and topographic variables on bacterial diversity is not clear. Please check this.
- Please improve figure 3, it is not possible to read the legend in the axes
- Please insert legend in tables and figures.
Author Response
Thank you for your comments, whcih we have uploaded response in word.

Reviewer 3 Report
Accept after minor revision.

Author Response
Thank you for your comments, and we have uploaded response in word.
